# The Effect of Photosensitizer Metalation Incorporated into Arene–Ruthenium Assemblies on Prostate Cancer

**DOI:** 10.3390/ijms241713614

**Published:** 2023-09-02

**Authors:** Lucie Paulus, Manuel Gallardo-Villagrán, Claire Carrion, Catherine Ouk, Frédérique Martin, Bruno Therrien, David Yannick Léger, Bertrand Liagre

**Affiliations:** 1Univ. Limoges, LABCiS, UR 22722, Faculté de Pharmacie, F-87000 Limoges, France; lucie.paulus@etu.unilim.fr (L.P.); villagran@outlook.com (M.G.-V.); frederique.martin@unilim.fr (F.M.); david.leger@unilim.fr (D.Y.L.); 2Institut de Chimie, Université de Neuchâtel, Avenue de Bellevaux 51, CH-2000 Neuchâtel, Switzerland; bruno.therrien@unine.ch; 3Univ. Limoges, CNRS, Inserm, CHU Limoges, BISCEm, UAR 2015, US 42, F-87000 Limoges, France; claire.carrion@unilim.fr (C.C.); catherine.ouk@unilim.fr (C.O.)

**Keywords:** prostate cancer, photodynamic therapy, photosensitizers, arene–ruthenium complexes, apoptosis

## Abstract

Prostate cancer is the second most common cancer for men and a major health issue. Despite treatments, a lot of side effects are observed. Photodynamic therapy is a non-invasive method that uses photosensitizers and light to induce cell death through the intramolecular generation of reactive oxygen species, having almost no side effects. However, some of the PSs used in PDT show inherent low solubility in biological media, and accordingly, functionalization or vectorization is needed to ensure internalization. To this end, we have used arene–ruthenium cages in order to deliver PSs to cancer cells. These metalla-assemblies can host PSs inside their cavity or be constructed with PS building blocks. In this study, we wanted to determine if the addition of metals (Mg, Co, Zn) in the center of these PSs plays a role. Our results show that most of the compounds induce cytotoxic effects on DU 145 and PC-3 human prostate cancer cells. Localization by fluorescence confirms the internalization of the assemblies in the cytoplasm. An analysis of apoptotic processes shows a cleavage of pro-caspase-3 and poly-ADP-ribose polymerase, thus leading to a strong induction of DNA fragmentation. Finally, the presence of metals in the PS decreases PDT’s effect and can even annihilate it.

## 1. Introduction

In 2020, prostate cancer (PCa) was the second most commonly occurring cancer in men and the fourth most common cancer overall. With more than 1.4 million new cases and 375,000 deaths recorded worldwide, PCa is the fifth most deadly cancer among men [1]. Following the detection, and depending on the grade and stage of the cancer, the patients are offered different treatments. If the cancer is detected early, the choice of treatment is oriented toward active surveillance or chemotherapy. Otherwise, for advanced cancer, a surgical approach with radical prostatectomy (partial or total removal of the prostate), which can be coupled to external radiotherapy or hormonal therapy, is preferred [2]. Today, those treatments coupled with early detection allow good management of the disease. Despite therapeutic advances, cancer relapse and many side effects such as pain and urinary or erectile dysfunctions are often observed [3,4]. Therefore, improving the patient’s quality of life while preserving the surrounding healthy tissues remains a priority. For these reasons, photodynamic therapy (PDT) could play an important role, thanks to its minimal invasiveness and high precision of treatment [5,6,7]. 

Indeed, PDT appears to be a promising alternative, showing a reduction in side effects during clinical trials [8,9]. The molecular mechanism of PDT resides in the interaction between three main actors: a photosensitizer (PS), a light source with an appropriate wavelength to excite the PS, and molecular oxygen. In PDT, the PS is activated by light, reaching an excited singlet state (S_1_). In that state, the PS is very unstable and loses its excess energy as it returns to its ground state (S_0_) or to a triplet state (T_1_). During this long-lived excited triplet state, the PS gradually returns to the ground state through type I or type II photochemical reactions. For type I reactions, the PS (T_1_) reacts with a biological substrate via hydrogen or electron transfer, thus leading to the production of free radical species. These species can react with O_2_ and induce superoxide anion (O_2_^•−^), hydroxyl radical formation (^•^OH), or hydrogen peroxide (H_2_O_2_) production. For type II reactions, the PS (T_1_) transfers its energy directly to O_2_, inducing singlet oxygen (^1^O_2_) production (Figure 1). Overall, PDT kills cells based on the generation of reactive oxygen species (ROS), which leads to cellular toxicity upon reaction with cellular molecules (lysosomes, DNA, mitochondria, etc.) [10]. Actually, tetrapyrrole compounds such as porphyrins, chlorins, bacteriochlorins, and phthalocyanines are the most commonly used PSs in PDT [11,12].

Despite all these benefits, PDT has some drawbacks, like the poor solubility of PSs in water and biological media, as well as PS aggregation and restricted tumor selectivity. Taken together, these drawbacks limit the use of standard PSs in clinical protocols. To increase the bioavailability of PSs, several new drug delivery systems are emerging through the vectorization of PSs in order to increase internalization into cells. PSs can be conjugated to polyamine, because tumor cells have a high production of polyamine and polyamine transport systems (PTS), leading to strong anticancer efficacy [13]. Nanoparticles (NPs) are also used to transport PSs in solid tumors through passive targeting and show a strong potential for clinical use. Nevertheless, functionalization and/or vectorization are often used to optimize water solubility and increase the internalization of PSs [14,15,16,17]. 

Another strategy to treat cancer consists of using metal-based therapeutic agents. Metals in bioinorganic chemistry can trigger different mechanisms, modify toxicity, and provide structural diversity. Ruthenium (Ru) anticancer agents, in particular, which are less toxic than other heavy metals (like platinum) [18], are receiving a lot of attention. The application of Ru is not new since its toxicity, anticancer, antimetastatic, and antiangiogenic properties have already been highlighted [19,20]. Moreover, Ru can bind to transferrin receptors in order to enter cells, and because cancer cells contain a high transferrin receptor density, Ru accumulates preferentially into tumors [21]. Therefore, it is not surprising that Ru and PSs have been combined for PDT applications.

Merging Ru and PSs can be carried out in different ways. The metal can be introduced at the core of the photosensitizer [22] or at its periphery [23].

Another way to combine Ru and PSs is to prepare water-soluble organometallic metallacages as hosts to deliver the PS to cancer cells or to use multidentate PSs to construct Ru-based assemblies [24]. In our study, two different types of arene Ru assemblies as carriers for PSs were prepared. One is a prismatic metallacage (**C3** Figure 2), which can host a PS as a guest in its internal cavity, transporting and releasing the PS into the cells, while the others are cubic metalla-assemblies (**C1** and **C2** Figure 3), in which two PSs are part of the structure [25,26].

In addition, the influence of the metalation of the PS on the PDT activity of these metalla-assemblies was studied by introducing diamagnetic metals (Zn^2+^ or Mg^2+^) or a paramagnetic metal (Co^2+^) at the core of the PS (Figure 3).

First of all, we tested the anticancer efficacy of all compounds on two human prostate cancer cell lines (DU 145 and PC-3) before investigating if metalation led to a decrease or an increase in photoactivity. We observed the strong anticancer efficacy of all compounds except for one. We also found that the addition of metals in the PS does not increase efficiency but rather reduces the photoactivity. Because PDT is based on ROS production, we demonstrated anticancer efficiency through ROS production only after the photoactivation of the compounds. Moreover, to better understand the cell death process involved, we analyzed the apoptotic pathway leading to anticancer efficiency.

## 2. Results

### 2.1. Cytotoxic and Phototoxic Effects 

In order to determine the phototoxicity of our compounds in vitro, two human PCa cell lines (DU 145 and PC-3) were used. Cells were exposed or not to PDT with red irradiation and phototoxic effects were determined 24h post-PDT using an MTT assay. All of the seven compounds tested had no toxic effects in the dark on both cell lines (Figure 4 and Appendix A). The IC_50_ values were calculated for all compounds (Figure 4 and Appendix A). We also evaluated the effectiveness of our compounds by determining the photoindex (PI) of the compounds (IC_50_ without irradiation/IC_50_ with irradiation) (Table 1). The IC_50_ without irradiation was not reached at the concentrations tested, except for **Mg-Por-C3** in DU145 cells. Almost all of the compounds induced a strong decrease in cell viability in a dose-dependent manner when irradiated, except **Co-TPyP-C2**, which showed no effect on both cell lines even at 4 µM (Figure 4).

We observed that **TPyP-C1** was much more efficient than **Zn-TPyP-C1**, particularly for DU 145 cells, with an IC_50_ of 14 nM compared to 302 nM, respectively, and showing a very good PI of >7. Similar results were observed in PC-3 cells with respective IC_50_ values of 53 and 300 nM. For **C2**, similar results were observed, with a better efficiency of **TPyP-C2** than **Zn-TPyP-C2** in DU 145 cells (13 nM and 252 nM) with a PI of >7 for **TPyP-C2**. Alike results were observed for PC-3 (28 nM and 332 nM) with a PI of >3 for **TPyP-C2**. Regarding **C3**, **Por-C3** demonstrated a much better efficiency than **Mg-Por-C3**. Indeed, in DU145 cells, the IC_50_ for **Por-C3** was 504 nM compared to 1368 nM for **Mg-Por-C3**, while on PC-3 cells, the IC_50_ was 633 nM and 2181 nM, respectively.

Taken together, these results indicate that the presence of two PSs per metallacage can reduce the dose necessary for PDT activity (**C1** and **C2** compared to **C3**) (Appendix A). Furthermore, we proved that metal-free PSs (**TPyP-C1, TpyP-C2**, and **Por-C3**) are far more effective than compounds with metalated PSs (**Zn-TPyP-C1, Zn-TpyP-C2**, and **Mg-Por-C3**). 

For most of the following experiments, compounds were used at the IC_50_ values determined upon irradiation.

### 2.2. ROS Production 

Cell death through PDT generally occurs via the generation of intracellular ROS. Therefore, intracellular ROS levels using DCFDA staining after PDT were measured. Flow cytometry analyses show that after photoactivation, cells exposed to the compounds have enhanced intracellular ROS levels. Moreover, a greater ROS production in **TPyP-C1**+light than **Zn-TPyP-C1**+light is observed in both cell lines, suggesting that metalation reduces ROS production (Appendix A). When looking at **C2**, the results are different. Indeed, there is no significant difference between **TPyP-C2** and **Zn-TPyP-C2** in ROS production. There is still a difference between **TPyP-C2**+light and **Zn-TPyP-C2**+light compared to **Co-TPyP-C2**+light, which shows worse ROS production: 45% of positive gated in PC-3 cells (Figure 5). Regarding the third cage (**C3**), **Mg-Por-C3**+light leads to a decrease in ROS production compared to **Por-C3**+light in PC-3 cells (41% compared to 88%) (Appendix A).

Those results coincide with the phototoxicity of our compounds, in fact, metalation leads to decreased ROS production, especially for **Zn-TPyP-C1, Co-TpyP-C2**, and **Mg-Por-C3** (on PC-3 cells only). A shorter distance between the two PSs leads to a decrease in cytotoxicity, but no change in ROS production. The same results were observed in the DU 145 cell line except for **Mg-Por-C3**+light, which showed no significant difference from **Por-C3**+light.

### 2.3. Confocal Microscopy Analysis

For confocal microscopy analysis, the compounds were used at a concentration near the IC_50_ observed after irradiation. All compounds possess natural fluorescence in the red or infrared region [25,26,27]. Red fluorescence was clearly observed in the cytoplasm, indicating cellular internalization of the compounds in both cell lines except for **Co-TPyP-C2** (Figure 6). In order to see if compounds co-localized with organelles, cells were also co-treated with LysoTracker, MitoTracker, or EndoplasmicReticulum-Tracker (ER-Tracker). The results show that **TPyP-C2** and **Zn-TPyP-C2** do not co-localize with organelles (no yellow fluorescence) (Figure 6). Similar results were observed for other compounds (Appendix A) and they seem to be localized only in the cytoplasm, while **Co-TPyP-C2** was not internalized at all. These data suggest that all compounds (except **Co-TPyP-C2**) were taken up by both cell lines with excellent internalization.

### 2.4. Apoptosis and DNA Fragmentation

Because apoptosis is often involved in PDT treatments, we evaluated apoptosis effects at higher concentrations (10 times the IC_50_ for **TPyP-C1 or C2**, 8 times the IC_50_ for **Por-C3 and Zn-TpyP-C2**, and 3 or 2 times the IC_50_ for **Zn-TpyP-C1** and **Mg-Por-C3**, respectively) in order to have a better response. Western blotting was performed on apoptotic-related proteins pro-caspase-3 and its cleaved form as well as native poly-ADP-ribose polymerase (PARP-1) and its cleaved form. In the case of PARP-1, it is involved in the late apoptosis pathway.

First, for the **C1** series, the results show that there was no significant decrease in pro-caspase-3 on the PC-3 cell line for **TPyP-C1** and no cleaved caspase-3. However, **TPyP-C1** induced a major cleavage of PARP-1 at 9.5-fold at a higher concentration (150 nM) after photoactivation. Similar results were observed for **Zn-TPyP-C1** at 2-fold only for cleaved PARP-1 on PC-3 cells. On the other hand, on DU 145 cells, a strong cleavage of pro-caspase-3 was spotted for **Zn-TPyP-C1** (77-fold) at a very high concentration (1000 nM). Likewise, PARP-1 cleavage was higher for **Zn-TPyP-C1** at 30-fold compared to 5-fold for **TPyP-C1** at 1000 nM (Appendix A). 

Next, for the **C2** group, **TPyP-C2** and **Zn-TPyP-C2** were identified as good inducers of apoptosis at higher concentrations through PARP-1 cleavage at 5.4-fold and 4.6-fold, respectively (Figure 7A). Again, on DU 145 cells, a very strong cleavage of pro-caspase-3 was observed at 9-fold for **TPyP-C2** using a higher concentration related to the cleavage of PARP-1 at 21-fold for **TPyP-C2** and a lower cleavage for **Zn-TPyP-C2** (Figure 7B). As expected, and unlike the other compounds, **Co-TPyP-C2** showed no evidence of apoptosis.

Finally, for the **C3** group, on the PC-3 cell line, we saw no change in pro-caspase 3 cleavage but a good cleavage of native PARP-1, mainly for **Por-C3** after photoactivation, at 10-fold for cleaved PARP-1 at IC_50_. For the DU 145 cells, we observed a good cleavage of pro-caspase 3 for **Por-C3** only after photoactivation. As seen in Figure 7, a strong cleavage of PARP-1 is noticeable for **Por-C3** at 54-fold and 65-fold at IC_50_ and a higher concentration, respectively (Appendix A). 

Furthermore, to confirm apoptosis and in order to see nuclear changes, DNA fragmentation was performed by ELISA. Overall, all compounds induced an increase in DNA fragmentation except for **Co-TPyP-C2**. In group **C1** on PC-3 cells, **TPyP-C1** showed a strong increase in DNA fragmentation at 8.67-fold compared to the other derivatives. Indeed, in the presence of Zn, DNA fragmentation still occurs, but to a lesser extent (3-fold), while no significant differences were observed on DU 145 cells (Appendix A). If we now look at the group **C2** for PC-3 cells, the results are different. In fact, **TPyP-C2** shows a DNA fragmentation (4-fold) that is much lower than **Zn-TPyP-C2** by almost 10-fold. On DU 145 cells, **TPyP-C2** displays a stronger DNA fragmentation at 4.5-fold compared to **Zn-TPyP-C2** (2-fold) (Figure 8). As for **Co-TPyP-C2**, no differences are observed on both cell lines. Then, for the **C3** group, **Por-C3** induced a much better fragmentation on PC-3 cells and DU 145 at 8-fold and 3.8-fold, respectively, compared to **Mg-Por-C3** at 5.5-fold and 2.3-fold (Appendix A). 

Taken together, these results suggest that all compounds (except **Co-TPyP-C2**) induce apoptosis and DNA fragmentation. It is important to highlight that, in general, the metalation (Zn, Mg) of the PS reduces the cleavage of the various proteins involved in the apoptotic pathway and DNA fragmentations.

## 3. Discussion

In this study, we have evaluated the anticancer potential of Ru assemblies containing metal-free and metalated photosensitizers on prostate cancer cell lines (PC-3 and DU 145). First, we evaluated the phototoxicity of the compounds and we highlighted their significant cytotoxic effects with IC_50_ values in the nanomolar range. Compounds built with two PSs (**C1** and **C2**) have a stronger effect than those with the PS inside the cavity (**C3**). Prior to our study, it had already been demonstrated that the release of porphin from cubic or prismatic cages was possible [27]. Moreover, once inside the cells (HeLa, Me300, A2780, A2780cisR, and A549), the PS was released from the cage and irradiated [28] without attempting to locate the compounds. To explain the release of the PS from the metallacages, two mechanisms have been suggested: through the opening of cages or from a rupture of the cage (partial or total) [27,28].

The main objective was to determine the difference in efficiency in the presence or absence of metals (zinc, cobalt) inside the PS in the center of the tetrapyridylporphyin panels or as a guest inside the cavity of the prism (magnesium). According to our results, the presence of metal decreases PDT efficiency and annihilates it in the case of Co. Both **TPyP-C1** and **TPyP-C2** (the metal-free derivative) have higher phototoxicity than their Zn or Co analogs. One possible explanation is the fluorescence emission of the PS. Indeed, the fluorescence emission of PSs containing Zn or Mg as a metal center is higher than metal-free analogs [25,26]. In addition, metal coordination in porphyrins increases the rate of decay for the intersystem crossing to the triplet state (Figure 1), resulting in a decrease in the fluorescence quantum yield [29]. **Co-TPyP-C2** did not show any phototoxic activity, as expected for a Co(II) porphyrin derivative, due to the absence of a triplet state [30] and also a lack of internalization. Another study goes along the same lines, in which paramagnetic species result in a reduction in the lifetime of the triplet state. Concerning the diamagnetic ion (Zn), a similar conclusion was made, as the triplet decay rates are about four times greater than metal-free porphyrins [31]. Taking this into account, a lower production of ROS is expected. Contradictory results can be explained by the formation of free radical species (Type I reaction) when using diamagnetic metals, while complexes without metal tend to produce singlet oxygen (Type II reaction).

We have found that all compounds localized into the cytoplasm, except for **Co-TPyP-C2**. A confocal microscopy study demonstrated that Zinc (II) phthalocyanine co-localized with the Golgi apparatus in most cases and with mitochondria only after prolongated incubation [32]. Although we are not working with phthalocyanine in our study, it is possible that our compounds act in the same way. Furthermore, the study also highlighted that 2 h incubation led to cell death through necrosis, and 24 h incubation mainly led to apoptosis, showing that it is possible in vitro to modulate cell death. Even if our compounds did not co-localize with mitochondria after prolongated incubation, it is possible that our compounds behave in the same way and could lead to necrosis, too. If so, it would be interesting to see if they co-localized with the Golgi apparatus. A previous study demonstrated that diamagnetic metals such as Zn, Pd, In, Sn, or Lu co-coordinated with the tetrapyrrole nucleus, allowing photosensitizing activity, while paramagnetic ones (Fe, Cu, Co) did not [33] in the case of phthalocyanines. In this article, we have proved that this applies to porphyrin as well. 

In the dark, our compounds with the addition of metal (in the center of the PS or inside the cavity) show a greater cytotoxicity. These results correlate with an earlier study in which it was reported that the presence of Zn exhibits some toxicity in the dark at high concentrations on synovial cells [26]. Furthermore, the regression of cell growth has been attributed to the presence of Zn inside porphyrins [34,35]. Regarding the results for the apoptosis pathway, PDT is well known to lead to apoptosis. With this in mind, we validated that most of our compounds induced apoptosis through pro-caspase 3 and PARP-1 cleavage, leading to DNA fragmentation. Once again, these results are consistent with the literature. Indeed, it has already been demonstrated that fluorinated Ru porphyrin has DNA interaction leading to its cleavage into melanoma cells [36].

Future research to determine the mechanism that some of these systems follow, once cellular internalization has taken place, could be useful. Indeed, experiments on the production of singlet oxygen could thus be carried out under optimal and reliable experimental conditions. In addition, a study of other cell death pathways (necrosis, necroptosis, etc.) should be carried out in view of the results observed. The in vivo use of the various compounds would be the ultimate goal of this study. However, one possible limitation would be the cytotoxicity of Ru at higher concentrations. 

## 4. Materials and Methods

### 4.1. Materials

RPMI 1640 medium, RPMI red-phenol-free medium, fetal bovine serum (FBS), L-glutamine, and penicillin–streptomycin were purchased from Gibco BRL (Cergy-Pontoise, France). 3-(4,5-dimethylthiazol-2-yl)-2,5-diphenyltetrazoliumbromide (MTT), human anti-β-actin antibody, cell death detection enzyme-linked immunosorbent assay^PLUS^ (ELISA), and 2′,7′-dichlorofluorescéine diacetate (DCFDA) were obtained from Sigma-Aldrich (Saint-Quentin-Fallavier, France). LysoTracker, goat anti-rabbit IgG H&L horseradish peroxidase (HRP) secondary antibody, Poly-ADP-ribose polymerase (PARP) antibody, caspase-3 antibody, and cleaved caspase-3 antibody were purchased from Cell Signaling Technology—Ozyme (Saint-Quentin-en-Yvelines, France). MitoTracker, ER-Tracker, and rabbit anti-mouse IgG-IgM H&L HRP secondary antibody were obtained from Invitrogen—Thermo Fisher Scientific (Villebon-sur-Yvette, France). Immobilon Western Chemiluminescent HRP Substrate was acquired from Merck (Lyon, France).

### 4.2. Synthesis of Compounds

#### 4.2.1. Photosensitizers 

5,10,15,20-tetra(pyridyl-4-yl)-21*H*,23*H*-porphine (**TPyP**) was purchased from Sigma-Aldrich, while Zn(II)-5,10,15,20-tetra(pyridyl-4-yl)-21*H*,23*H*-porphine (**Zn-TPyP**) and Co(II)-5,10,15,20-tetra(pyridyl-4-yl)-21H,23H-porphine (**Co-TPyP**) were obtained from Porphychem (Dijon, France). The other photosensitizers, 21*H*,23*H*-porphine (**Por**), and Mg(II)-porphine (**Mg-Por**), were synthesized according to the literature [37] (Figure 3). 

#### 4.2.2. Cages 

The metallacages, [Ru_8_(η^6^-p-^i^PrC_6_H_4_Me)_8_(μ^4^-H_2_-TPyP-κN)_2_(μ-C_6_H_2_O_4_-κO)_4_][CF_3_SO_3_]_8_ (**TPyP-C1**), [Ru_8_(η^6^-p-^i^PrC_6_H_4_Me)_8_(μ^4^-Zn-TPyP-κN)_2_(μ-C_6_H_2_O_4_-κO)_4_][CF_3_SO_3_]_8_ (**Zn-TPyP-C1),** [Ru_8_(η^6^-p-^i^PrC_6_H_4_Me)_8_(μ^4^-H_2_-TPyP-κN)_2_(μ-C_2_O_4_-κO)_4_][CF_3_SO_3_]_8_ (**TPyP-C2**), [Ru_8_(η^6^-p-^i^PrC_6_H_4_Me)_8_(μ^4^-Zn-TPyP-κN)_2_(μ-C_2_O_4_-κO)_4_][CF_3_SO_3_]_8_ (**Zn-TPyP-C2**), and [Ru_8_(η^6^-p-^i^PrC_6_H_4_Me)_8_(μ^4^-Co-TPyP-κN)_2_(μ-C_2_O_4_-κO)_4_][CF_3_SO_3_]_8_ (**Co-TPyP-C2**) were synthesized as reported in the literature [25,38]. Regarding {[(ր^6^-p-cymene)_6_Ru_6_(2,5-dioxydo-1,4-benzoquinonato)_3_(2,4,6-tri(pyridin-4-yl)-1,3,5-triazine)_2_][SO_3_CF_3_]_6_⊂Porphine or Mg-Porphine}, (**Por-C3**) or (**Mg-Por-C3**) were synthesized as reported in the literature [26,27] (Figure 2).

### 4.3. Cell Culture and Treatment 

Prostate cancer cell lines (PC-3 and DU 145) were purchased from the American Type Culture Collection (ATCC) (LGC Standards, Middlesex, UK). Cells were grown in RPMI 1640 medium supplemented with 10% fetal bovine serum, 1% L-glutamine, 100 U/mL penicillin, and 100 µg/mL streptomycin. Cultures were maintained in a humidified atmosphere with 5% CO_2_ at 37 °C. For all experiments, cells were seeded at 1.8 × 10^4^ cells/cm^2^ for DU 145 and PC-3 cells. Stock solutions of all compounds were prepared in DMSO (1 mM) and prior to use were diluted in a culture medium to obtain the appropriate final concentrations. The concentration of DMSO was never more than 0.4% in the cell medium.

### 4.4. In Vitro Protocol of PDT

Prostate cancer cells were seeded in 25 cm^2^ or 96-well (6000 cells per well) culture plates and were grown for 24 h in a culture medium prior to exposure or not to compounds. After 24 h, the culture medium was replaced by a red-phenol-free culture medium before PDT. Then cells were irradiated or not with a 630–660 nm CURElight lamp at 75 J/cm^2^ (PhotoCure ASA, Oslo, Norway). At 24 or 48 h after irradiation, cells were removed for analysis.

### 4.5. In Vitro Phototoxicity of Compounds 

Antiproliferative assays were determined using an MTT assay. Cells were seeded in 96-well culture plates and treated as described above with the compounds. After 24 h of incubation, cells were irradiated or not. MTT assays were performed 24 and 48 h after irradiation and cell viability was expressed as a percentage of each treatment condition by normalizing to untreated cells.

### 4.6. Intracellular ROS Generation 

ROS generation was quantified using a detection assay kit that uses the cell-permeant reagent 2′,7′-dichlorofluorescein diacetate (DCFDA). The cells were seeded in 25 cm^2^ tissue culture flasks and were grown 24 h prior to exposure or not to compounds at respective IC_50_ values. Then, the cells were stained with DCFDA for 30 min at 37 °C. After washing, cells were irradiated or not. ROS generation was examined by flow cytometry immediately after PDT. H_2_O_2_ was used as a positive control at 800 µM.

### 4.7. Localization 

To determine compound localization, cells were seeded in lab tek chamber slides and were grown for 24 h prior to exposure to compounds at IC_50_ values. After 24 h incubation, cells were co-treated at 37 °C with MitoTracker (50 nM), ER-Tracker (500 nM), or LysoTracker (50 nM) for 45 min, 30 min, and extemporaneously, respectively. Compound localization was determined by confocal microscopy using compound fluorescence with LysoTracker fluorescence (excitation/emission: 504/511 nm), MitoTracker fluorescence (excitation/emission: 490/516 nm), and ER-Tracker fluorescence (excitation/emission 504/511 nm). Photos were taken with a confocal microscope (laser Zeiss LSM 510 Meta—×1000).

### 4.8. In Vitro Apoptosis Detection

Cells were seeded in 25 cm^2^ tissue culture flasks and were grown for 24 h prior to exposure or not to compounds at IC_50_ values and were irradiated or not. At 24 h post-PDT, cells were recovered and divided into two groups. For the first group, cells were lysed in the RIPA lysis buffer. Protein levels were determined using the Bradford method. Western blotting was performed on apoptosis-related proteins, human anti-poly-ADP-ribose polymerase (PARP-1) (1:1000), human anti-caspase 3 (1:1000), human anti-pro caspase 3 (1:1000). Human anti-β-actin (1:5000) was used as a loading control. After incubation with the appropriate secondary antibodies, blots were developed using Immobilon Western Chemiluminescent HRP Substrate and a G:BOX system (Syngene, Cambridge, UK). 

The other group was used to assess DNA fragmentation. The Cell Death ELISA^PLUS^ kit was used, allowing for the specific determination of mono- and oligonucleosomes in the cytoplasmic fraction of cell lysates. Cytosolic extracts were obtained according to the manufacturer’s protocol and apoptosis was measured as previously described [13]. DNA fragmentation was measured and the results were reported as n-fold compared to control.

### 4.9. Statistical Analysis

All data are expressed as the mean ± standard error of the mean (SEM) of separate experiments. The statistical significance of results was evaluated by a two-tailed unpaired Student’s *t*-test, as * *p* < 0.05, ** *p* < 0.01, and *** *p* < 0.001 or # *p* < 0.05, ## *p* < 0.01, and ### *p* < 0.001.

## 5. Conclusions

Herein, we evaluated for the first time the anticancer efficacy of **TPyP-C1, Zn-TPyP-C1, TPyP-C2, Zn-TPyP-C2, Co-TpPyP-C2, Por-C3**, and **Mg-Por-C3** in human prostate cancer cell lines. We can conclude that the intracellular accumulation and distribution of our compounds show a strong anticancer efficacy in vitro resulting in cell death via the apoptotic pathway. The addition of Zn^2+^ or Mg^2+^ inside the tetrapyrrole ring center decreases the anticancer effect. Furthermore, the addition of Co^2+^ leads to a total absence of effect, due to the paramagnetic character of cobalt and the absence of internalization. Nevertheless, most compounds show promising results. To complete this study, it would be useful to validate the two most effective derivatives (**TPyP-C1** and **TPyP-C2**) in vivo in order to validate our findings.

## Figures and Tables

**Figure 1 ijms-24-13614-f001:**
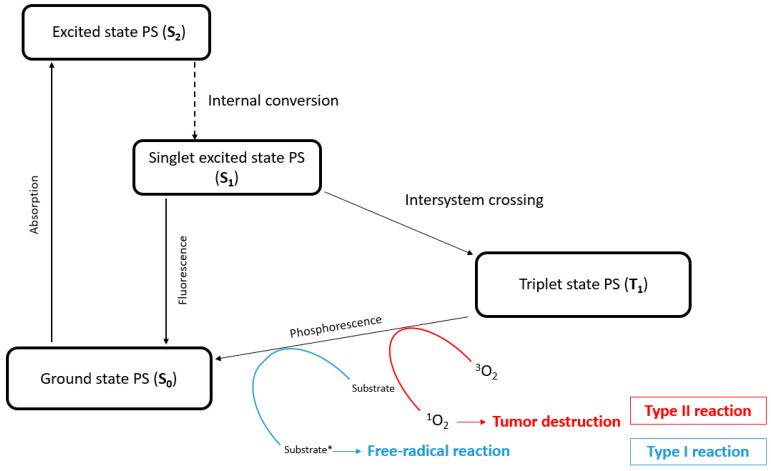
Representation of the Jablonski diagram. Upon light absorption, the PS moves from its ground state (S_0_) to an excited state (S_2_), then by internal conversion to a lower-energy singlet excited state (S_1_), and finally by intersystem transition to a triplet state (T_1_). Substrate*: substrate ions and radicals.

**Figure 2 ijms-24-13614-f002:**
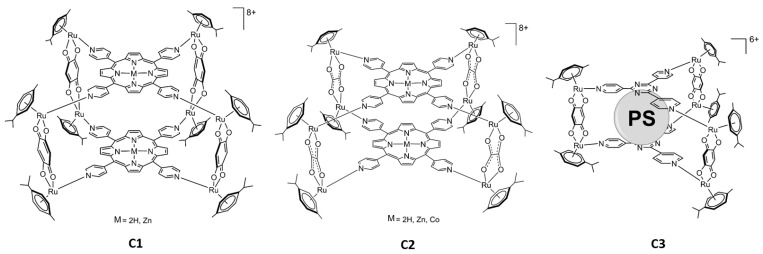
Structures of the Ru metalla-assemblies used in this work. **TPyP** and **Zn-TPyP** in **C1** and **TPyP**, **Zn-TPyP,** and **Co-TPyP** in **C2**. Photosensitizers are represented by a sphere (PS). **Por** and **Mg-Por** were inserted into **C3**.

**Figure 3 ijms-24-13614-f003:**
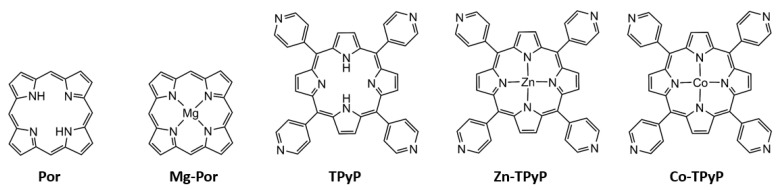
Photosensitizers used in this study. From left to right, 21H, 23H-porphine (**Por**), Mg(II)-porphine (**Mg-Por**), 5,10,15,20-tetra(pyridyl-4-yl)-21*H*,23*H*-porphine (**TPyP**), Zn(II)-5,10,15,20-tetra(pyridyl-4-yl)-21H,23H-porphine (**Zn-TPyP**), and Co(II)-5,10,15,20-tetra(pyridyl-4-yl)-21*H*,23*H*-porphine (**Co-TPyP**).

**Figure 4 ijms-24-13614-f004:**
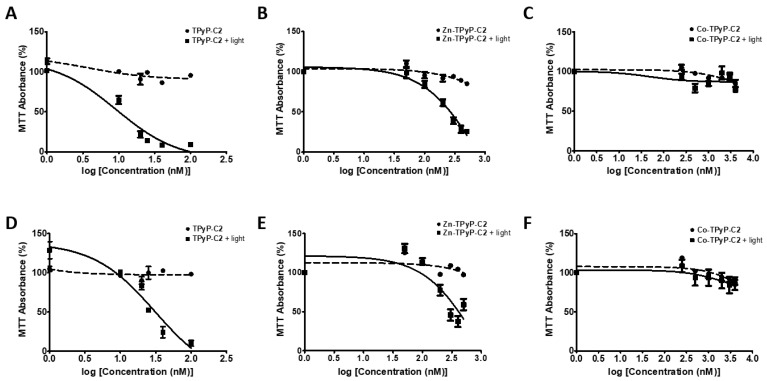
Phototoxicity of PSs on human prostate cancer cell lines. Cells were cultured in RPMI medium for 24 h. DU 145 were treated or not with compounds (**A**) **TPyP-C2**, (**B**) **Zn-TPyP-C2**, (**C**) **Co-TPyP-C2** and PC-3, (**D**) **TPyP-C2**, (**E**) **Zn-TPyP-C2**, or (**F**) **Co-TPyP-C2** for 24 h. Cells were irradiated (630 nm, 75 J/cm^2^) or kept in the dark. Compound toxicity at 24 h was followed by an MTT assay and IC_50_ values were calculated. Data are shown as mean ± SEM (n = 3).

**Figure 5 ijms-24-13614-f005:**
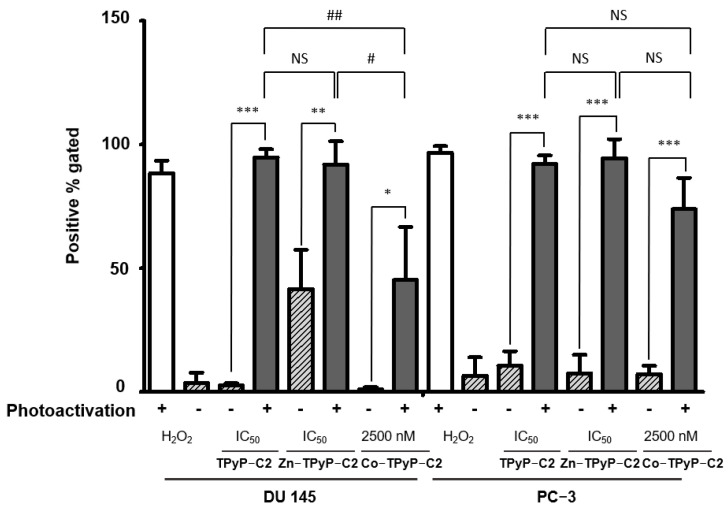
Induced ROS production in human prostate cancer cell lines PC-3 (right) and DU 145 (left). Cells were treated with compounds and photoactivated or not. Intracellular ROS levels using DCFDA staining were measured directly after PDT by flow cytometry. A higher fluorescence intensity resulting from a higher amount of 2′,7′-dichlorofluorescein (DCF) formation results in a shift to the right. Data are shown as mean ± SEM (n = 3). * *p* < 0.05; ** *p* < 0.01 and *** *p* < 0.001 relative to H_2_O_2_ or *# p* < 0.05; *## p* < 0.01 relative to compounds; NS: not significant.

**Figure 6 ijms-24-13614-f006:**
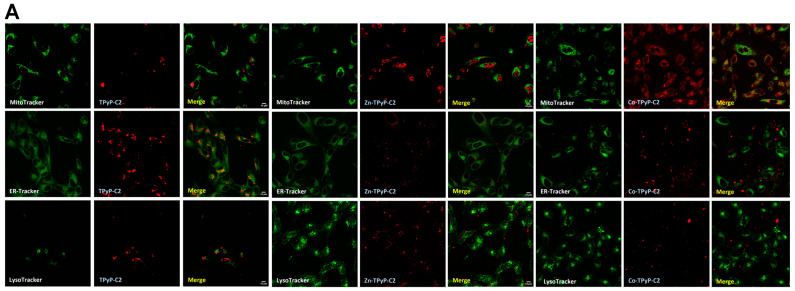
Localization of the **C2** series on human prostate cancer cells. DU 145 (**A**) and PC-3 (**B**) were grown for 24 h prior to exposure to **TPyP-C2**, **Zn-TpyP-C2**, or **Co-TPyP-C2** at IC_50_ values. After 24 hb cells were treated with MitoTracker, ER-Tracker, or LysoTracker for 45, 30 min, and extemporaneously, respectively. Localization was studied by confocal microscopy and photos were taken with a confocal microscope (laser Zeiss LSM 510 Meta—×1000).

**Figure 7 ijms-24-13614-f007:**
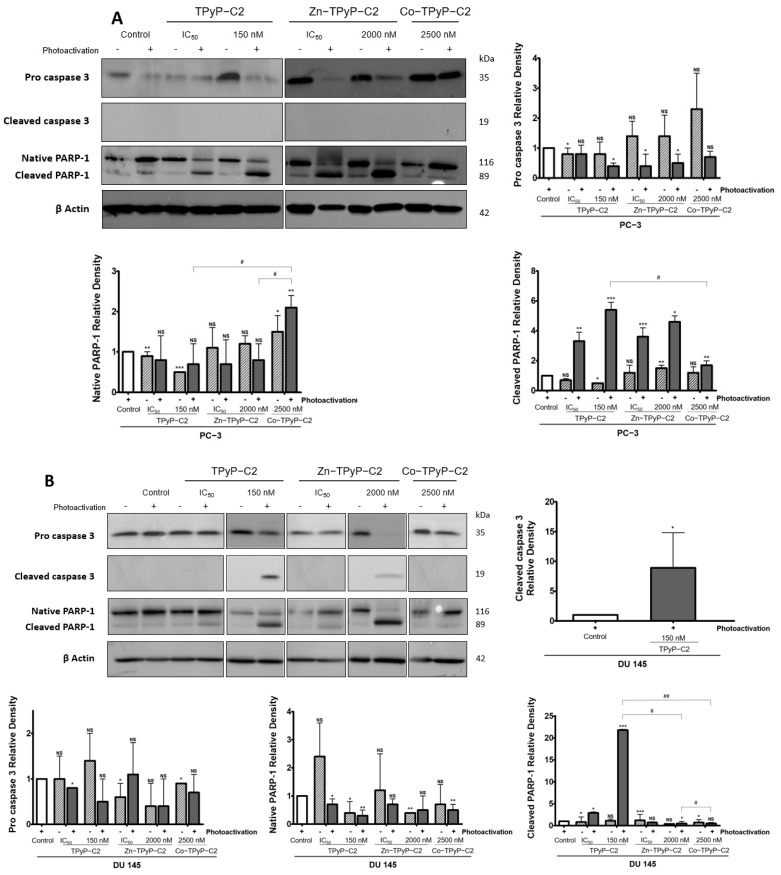
Effects of the **C2** compounds on human prostate cancer cell lines. PC3 cells (**A**) were treated or not with **TPyP-C2**, **Zn-TPyP-C2**, or **Co-TPyP-C2**. DU 145 cells (**B**) were treated or not with **TPyP-C2**, **Zn-TPyP-C2**, or **Co-TPyP-C2**. The expression of caspase-3 activation and PARP-1 cleavage was analyzed by Western blotting 24 h post-PDT. β-actin was used as a loading control. * *p* < 0.05, ** *p* < 0.01, and *** *p* < 0.001 relative to control group or *# p* < 0.05, *## p* < 0.01 relative to compounds; NS: not significant.

**Figure 8 ijms-24-13614-f008:**
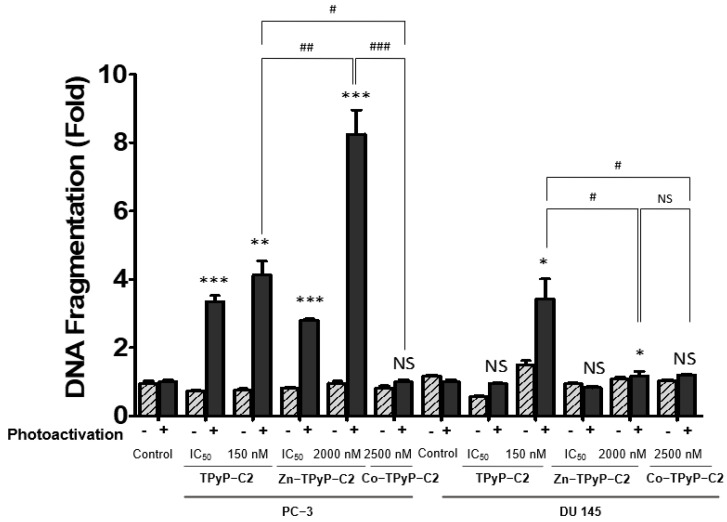
DNA fragmentation through ROS production in prostate cancer cell lines. DNA fragmentation in cells 24 h post-PDT was quantified from cytosol extracts by ELISA. Cells were treated with **TPyP-C2**, **Zn-TPyP-C2**, or **Co-TPyP-C2** on DU 145 and PC3 cell lines. The results are reported as n-fold compared to light control. Values are expressed as mean ± SEM (N = 3). * *p* < 0.05, ** *p* < 0.01, and *** *p* < 0.001 relative to control group or # *p* < 0.05; ## *p* < 0.01; ### *p* < 0.001 relative to compounds; NS: not significant.

**Table 1 ijms-24-13614-t001:** IC_50_ values (nM) determined with MTT assays on DU 145 and PC-3 cells. PI = IC_50_ without irradiation/IC_50_ with irradiation. Data are shown as mean ± SEM (n = 3). Not determined (n.d.).

	DU 145			PC-3		
Compounds	IC_50_ (nM) Light	IC_50_ (nM) Dark	PI	IC_50_ (nM) Light	IC_50_ (nM) Dark	PI
**TPyP-C1**	14 ± 4	>100	>7	53 ± 6	>100	>1
**Zn-TPyP-C1**	302 ± 29	>500	>1	300 ± 19	>500	>1
**TPyP-C2**	13 ± 2	>100	>7	28 ± 2	>100	>3
**Zn-TPyP-C2**	252 ± 31	>500	>1	332 ± 84	>500	>1
**Co-TPyP-C2**	n.d.	n.d.	n.d	n.d	n.d	n.d.
**Por-C3**	504 ± 44	>4000	>7	633 ± 88	>4000	>4
**Mg-Por-C3**	1368 ± 130	2133 ± 690	>2	2181 ± 460	>4000	=1.5

## Data Availability

Data sharing is not applicable.

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
