# Peer review of "The Effect of Photosensitizer Metalation Incorporated into Arene–Ruthenium Assemblies on Prostate Cancer"

_ijms, 2023, doi:10.3390/ijms241713614_

Round 1

Reviewer 1 Report

Essentially, the article is well-prepared, but it lacks some technical aspects.

Cytotoxicity graphs should be displayed for all compounds.

Could the authors present these charts using Hill curves?

Btw, how was IC50 determined? it is interesting since the cytotoxicity chart was created using the "connect the dots" method?

In the table in Figure 4 (tables and figures should be separated), IC50 values with and without illumination should be shown, along with the corresponding PI values.

Why only results obtained after 24h are presented since Authors declare that the measurements were carried out also after 48h?

The original histograms from the cytometer could support the results in Figure 5.

Given that the lamp spans the 630-660 nm range, could variations in the cytotoxicity of the examined compounds be attributed to disparities in their absorption peaks?

In light of the above comment, would it not be reasonable to conduct a measurement of singlet oxygen production?

How was the light power measured in the cell experiment?

Why was the amount of DMSO used as high as 0.4%? Does it not influence cell viability or expression of some important enzymes, as some authors mention e.g. doi: 10.17795/ajmb-33453?

Reviewer 2 Report

The manuscript proposes an interesting study on the effect of photosensitizers aimed to target prostate cancer.
The topic correlates to the journal.
Nevertheless, there are some minor issues that require to be addressed before proceeding with the publication, to enhance the quality and presentation to a broad audience.
The abstract reports a consistent summary of the article core, but it spans in a very scattered way so that the reader might be in trouble by trying to understand it. At this purpose, references list must be improved with additional, broader state-of-the-art sources, comparing different PDT agents and triggers for their delivery, in order to enhance the core discussion (e.g. doi: 10.3389/fchem.2020.573211), mainly in the introduction section and reduce the self-citation rate.
Moreover, aim and hypothesis are missing: it is suggested to structure them at the very end of the introduction, in order to create a straightforward flow throughout the article, for the readers’ benefit.
A ToC graphic would definitely help.
It is suggested to run an English check throughout the whole manuscript.
Check for typos.
Moreover, it is suggested to add on a list of abbreviations as a point-listed: it will boost the scientific appeal.  
It might help discuss over an additional section any possible limitations/future perspectives.

An English check would strongly boost the whole manuscript, which still reports on latin-based sentence structures.
